# Unforgettable Generalization in Language Models

**Eric Zhang, Leshem Choshen & Jacob Andreas**
MIT
{zeric,leshem,jda}@mit.edu

## Abstract

When language models (LMs) are trained to forget (or "unlearn") a skill, how precisely does their behavior change? We study the behavior of transformer LMs in which tasks have been forgotten via fine-tuning on randomized labels. Such LMs learn to generate near-random predictions for individual examples in the "training" set used for forgetting. Across tasks, however, LMs exhibit extreme variability in whether LM predictions change on examples *outside* the training set. In some tasks (like entailment classification), forgetting generalizes robustly, and causes models to produce uninformative predictions on new task instances; in other tasks (like physical commonsense reasoning and scientific question answering) forgetting affects only the training examples, and models continue to perform the "forgotten" task accurately even for examples very similar to those that appeared in the training set. Dataset difficulty is not predictive of whether a behavior can be forgotten; instead, generalization in forgetting is (weakly) predicted by the confidence of LMs' initial task predictions and the variability of LM representations of training data, with low confidence and low variability both associated with greater generalization. Perhaps most surprisingly, random-label forgetting appears to be somewhat insensitive to the contents of the training set: for example, models trained on science questions with random labels continue to answer other science questions accurately, but begin to produce random labels on entailment classification tasks. Finally, we show that even generalizable forgetting is shallow: linear probes trained on LMs' representations can still perform tasks reliably after forgetting. Our results highlight the difficulty and unpredictability of performing targeted skill removal from models via fine-tuning.

## 1 Introduction

In the modern approach to training language models (LMs), neural sequence models are first pre-trained on a large, minimally curated corpus (typically of web text), then fine-tuned with targeted demonstrations and human feedback. The LMs that result from this procedure often possess undesirable capabilities that creators do not wish to expose to users—for example, the ability to generate hate speech, or to answer questions about topics unrelated to the LM's target application. Can these capabilities be forgotten (or "unlearned")?

There has been widespread recent interest in developing and evaluating new techniques for removing both skills and declarative knowledge from LMs. This work has found that, on specific inputs of interest, LM behavior can be changed in targeted ways. But there has been comparatively little evaluation of *generalization* in forgetting—when an LM is trained not to respond (or to respond uninformatively) to a particular input, how does its behavior change on other inputs?

This paper studies generalization behavior in forgetting. We focus on forgetting of skills (rather than knowledge) via fine-tuning on randomly labeled data for the target task—a simple, widely used, and often highly effective method for forgetting (see Liu et al., 2024 for a recent survey). Surprisingly, we find wide variability *across tasks* in the effectiveness and generalization of random-label forgetting. When fine-tuning on randomized responses, models will change their behavior on training inputs, but sometimes do not change their

behavior at all for other instances of the same task—even when fine-tuning on accurate labels *does* lead to generalized improvements in accuracy. In additional experiments characterizing generalization in forgetting, we find:

1. The degree of forgetting is largely determined by the tasks that LMs are evaluated on, not the task LMs are trained to forget.

2. Generalization in forgetting is not determined by the difficulty of the task.

3. Properties that correlate with the generalization of forgetting include LM confidence as well as the variance of LM representations of training data.

4. Despite LMs' inability to respond correctly to prompts after applying this method, we are still able to recover the correct responses using linear probes. Hence, even successful forgetting is at best shallow, and does not remove information from LMs' representations.

Generalization of learning algorithms across problems and problem instances is a major focus of study in machine learning research. Our results show similarly complex, structured cross-task variability of generalization in forgetting, and underscore the need for additional research on the relationship between the training data used for forgetting and the effect of model predictions elsewhere.

## 2 Related work

Due to diverse privacy, security, and ethical concerns, machine unlearning has been conceptualized in many different ways. Early approaches defined unlearning as removing undesirable data from training sets (Cao & Yang, 2015; Bourtoule et al., 2021; Ginart et al., 2019). These approaches often require fundamental changes to model structure and/or training process, which is often infeasible.

Later work relaxed the requirement of removing data from the training set. Instead, models are required to behave similarly to models trained without undesirable data points, or are simply required to stop producing outputs with desirable features. Guo et al. (2020) develop a framework for linear classifiers, and Golatkar et al. (2020a) develop a method that scrubs information from linear probes. Neel et al. (2021); Sekhari et al. (2021); Thudi et al. (2022); Golatkar et al. (2020b; 2021); Mehta et al. (2022) and Chundawat et al. (2023) present theoretical frameworks for comparing an unlearned network to a fully-retrained networks, and they propose optimization-based methods to find unlearned network under additional assumptions like convexity. Foster et al. (2024) propose model editing techniques based on estimating parameter importance using fisher information. Kurmanji et al. (2023) distinguishe between different reasons for forgetting, arguing that distinct purposes like protecting user privacy, resolving confusion, and removing biases require distinct metrics. Graves et al. (2021) argue that selectively removing training data alone is insufficient, and propose a new threat model and techniques to address them.

For language models specifically, approaches to remove specific facts include gradient ascent on undesirable responses (Jang et al., 2023; Yao et al., 2023; Eldan & Russinovich, 2023), prompting with misinformation (Pawelczyk et al., 2023), linearly manipulating model representations (Ilharco et al., 2023; Belrose et al., 2023), non-linearly perturbing model representations (Li et al., 2024), and using new models to teach another model how to forget (Wang et al., 2023).

While some of this prior work has studied generalization (e.g. Li et al., 2024), they study a different kind of generalization: whether model behavior remains the same on non-targeted tasks. By contrast, our work focuses on generalization between instances of a single task.

Outside of research on unlearning, some past work has studied training on incorrect or random labels as a source of information about *learning* dynamics, for example finding that models often have similar embeddings (Morcos et al., 2018), learn in a similar order (Hacohen et al., 2020) and explaining the order of learning (Hacohen & Weinshall, 2022).

## 3 Experiment setup

**Method**  Our experiments in this paper study forgetting of capabilities (rather than factual knowledge). In order to enable uniform comparisons across tasks, we formulate each capability as binary multiple-choice question answering task. Each such task $T$ is associated with a training set $T_{\text{train}}$, a validation set $T_{\text{val}}$, and test set $T_{\text{test}}$. When studying forgetting, we first fine-tune the model on $T_{\text{train}}$ with early stopping performed by finding the checkpoint with the highest accuracy on $T_{\text{val}}$. Afterwards, we train the model to forget by fine-tuning the model again on $T_{\text{train}}$ but with labels chosen uniformly at random. This procedure is summarized in Figure 1.

**Quantifying forgetting**  We quantify forgetting with two metrics. The first is the gap between the accuracy after forgetting and the expected random accuracy (50% since the tasks are binary multiple choice), which we will call the **forget gap**:

$$\text{Forget Gap} = \text{Task Accuracy After Forgetting} - \frac{1}{2}$$

A gap of 0 indicates that the target task has been fully forgotten (all tasks involve a binary choice, and a random baseline obtains an accuracy of $\frac{1}{2}$). Larger values indicate that models still achieve non-trivial accuracy. We may also wish to interpret accuracy after forgetting relative to the upper bound provided by fine-tuning—an accuracy of 55% after forgetting might be interpreted as successful or unsuccessful if fine-tuned accuracy is 95% or 56%. To quantify this intuition, we define the **forget ratio**:

$$\text{Forget Ratio} = \frac{\text{Accuracy After Fine-Tuning} - \text{Accuracy After Forgetting}}{\text{Accuracy After Fine-Tuning} - \frac{1}{2}}$$

Here an forget ratio of 1 corresponds to complete forgetting, while a forget ratio of 0 corresponds to no decrease relative to the best attainable supervised performance.

**Tasks, evaluation details, and models**  We experiment on 21 multiple-choice tasks commonly found within the literature. **Commonsense Reasoning:** We evaluate PIQA (Bisk et al., 2020), ARC easy and challenge (Clark et al., 2018), and CREAK (Onoe et al., 2021). **Reading Comprehension:** We evaluate BoolQ (Clark et al., 2019), SciQ (Welbl et al., 2017), and PubMedQA (Jin et al., 2019). **Math:** We evaluate MathQA (Amini et al., 2019). **Toxicity:** We evaluate ToxiGen (Hartvigsen et al., 2022). **Entailment classification and other language understanding tasks:** We evaluate CoLA, MNLP, MRPC, QNLI, RTE, WNLI, CB, COPA, WIC and WSC (Wang et al., 2019). We selected these tasks to cover a broad spectrum of capabilities while also ensuring that they are multiple choice, which allows us to easily construct randomized alternatives for forgetting.

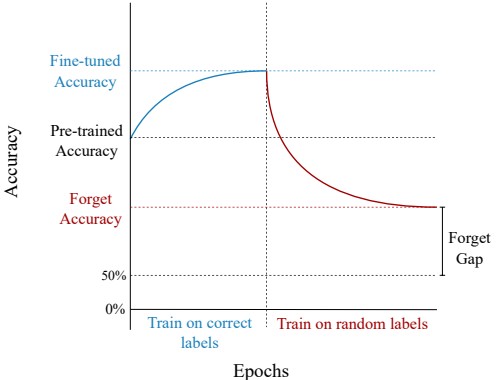

Figure 1: Stylized learning and forgetting curves. Our experiments first fine-tune a pre-trained LM, then train it further on random labels. We call the gap between the *forget accuracy* and the random chance accuracy (50%) the *forget gap*. In many tasks we find a nonzero forget gap: after training on random labels, LMs do not generalizably learn to produce random outputs on new task instances.

We follow the Language Model Evaluation Harness standards for 0-shot evaluation (Gao et al., 2023), including the default prompts and evaluation through probabilities of the choices. To facilitate comparison across tasks, we binarize the tasks by preserving two of the possible responses—the true response and one randomly chosen distractor—for each example. We evaluate models by picking the response with the highest average token likelihood and reporting the accuracy.

We use the publicly provided train, validation, and test sets. However, we found some datasets had train–test overlap. To decontaminate the datasets, we do not evaluate on questions that appeared in the training set. We also removed samples longer than 2048 characters in the prompt and combined response. Where validation sets do not exist, we use the test set. Unless otherwise specified, we limit each set to 1000 examples and subsample if needed, making training results more comparable and evaluation more efficient as proposed by Perlitz et al. (2023).

All experiments use Llama2 7-billion parameter base models (Touvron et al., 2023). Additional details may be found in Appendix A.

## 4 Does forgetting generalize?

Figure 2 summarizes the task accuracy without modification, after fine-tuning, and then after running our forgetting procedure. Test accuracy almost always increases after fine-tuning, although it could decrease slightly as the validation set is not identical to the test set. During the forgetting phase, however, we observe several distinct categories of behavior (1) forget accuracy is very similar to the fine-tuned accuracy, (2) forget accuracy decreases but is still above the pre-trained accuracy, and (3) forget accuracy decreases to below the pre-trained accuracy and possibly back to 50%. Case (2) is interesting because it demonstrates asymmetry between the learning and forgetting process, as the model is unable to forget what is has just learned (analogous to hysteresis in physical systems; Ewing, 1882).

Overall, we find that random-label forgetting often fails to *generalizably* remove the target behavior, but with wide variability across tasks. In general, tasks involving commonsense knowledge reasoning tasks are more resilient to forgetting, whereas lower-level linguistic acceptability and entailment classification tasks are more effectively forgettable.

We also examine cross-task forgetting, where we fine-tune the model on random labels from the training set of one task and then evaluate the model on the test set of another task. As shown in Figure 3, we find that the effectiveness of the forgetting procedure is largely determined by the tasks that the model is evaluated on—not the training task. Another surprising observation is that many tasks are more effectively forgotten when training on randomized labels of other tasks than from training on their own randomized labels. As observed in the individual task evaluation, GLUE tasks focused on specific capabilities are again more susceptible to forgetting in general, whereas commonsense reasoning tasks are more resilient to forgetting. Training on forgetting commonsense reasoning tasks are also generally more effective at triggering forgetting for other tasks.

## 5 When does generalization occur?

**Does forgetting require more examples?** We rule out the number of examples as the main explanation to forgetting generalization. For example, a possible concern could be that forgetting does not generalize because there are not enough training examples. We ran the same experiment with 100 examples of each task as well as 1000 (above). We find that despite an order of magnitude change, the level of forgetting is similar in both cases.

**Does forgetting occur with other methods?** To rule out the possibility that forgetting fails to generalize due to our method of training on randomized labels, we run another experiment where we train on flipped labels instead of randomized labels. The analysis is the same as before, except now we compute the forget ratio as:

$$\text{Forget Ratio} = \frac{\text{Accuracy After Fine-Tuning} - \text{Accuracy After Forgetting}}{\text{Accuracy After Fine-Tuning} - (1 - \text{Accuracy After Fine-Tuning})}$$

since we assume the minimum accuracy achievable should be $1 - $ Accuracy After Fine-Tuning.

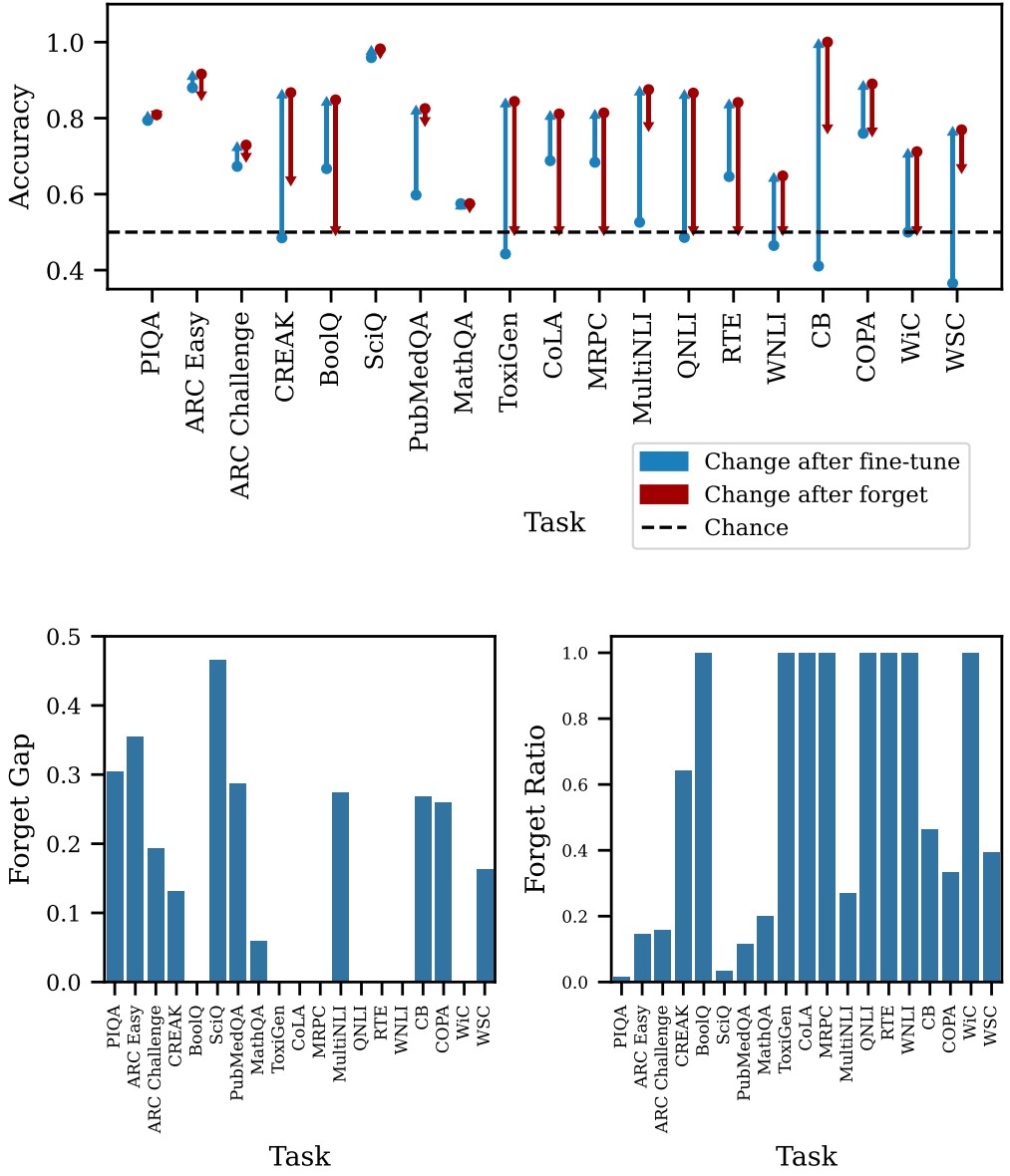

Figure 2: Single task forgetting. *Top*: The blue arrow visualizes the change in held-out accuracy after fine-tuning and the red arrow illustrates the change in accuracy after forgetting. We find that many tasks do not return to the expected accuracy of 50% after forgetting. *Bottom left*: The forget gap (difference between forgetting accuracy and the expected random accuracy of 1/2) across tasks. Smaller values correspond to a greater degree of forgetting. *Bottom right*: The forget ratio (the difference fine-tuned accuracy and the forget accuracy over the difference between fine-tuned accuracy and the expected random accuracy of 1/2). Larger forget ratios correspond to more successful forgetting.

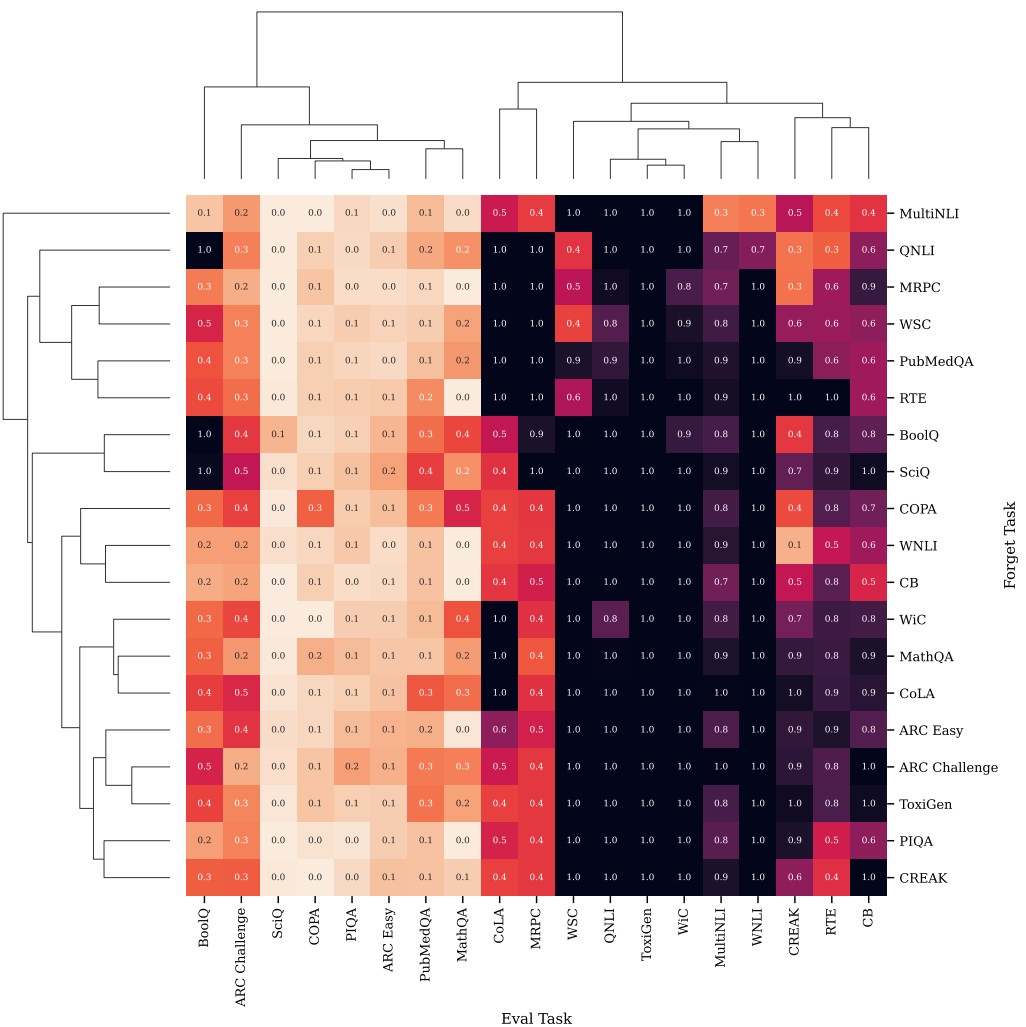

Figure 3: Cross-task forgetting (higher values indicate more successful forgetting). We fine-tune the model on random labels from one task and then evaluate the model on another task. The vertical axis displays the task the model was trained to forget and the horizontal axis displays the task the model was evaluated on. Surprisingly, certain capabilities are robust to forgetting even after fine-tuning on random labels. Moreover, the effectiveness of the forgetting procedure is largely determined by the tasks that the model is evaluated on, not the tasks that the model was trained to forget. Note that rows and columns are presented in different orders, and clustered using the UPGMA algorithm (Sokal & Michener, 1958)

As shown in Figure 8, the trends are the same. The same tasks that are robust to fine-tuning on randomized labels continue to be robust on fine-tuning on flipped labels. Thus, our results are likely not specific to the choice of randomized labels, but rather a property of how fine-tuning and the tasks interact.

**Does forgetting occur in other models?**   To understanding whether this forgetting behavior is unique to the LLama2 7-billion parameter model or to language models in general, we also experiment with GPT-J-6B, which is a slightly weaker model than the LLaMA-2-7B, and GPT-2, which is a significantly smaller model with 124M parameters (98% smaller).

As shown in Figure 8, while GPT-J and GPT-2 have lower fine-tuned accuracy, the forgetting ratio trends are broadly the same. Thus, the behavior is not unique to LLaMA-2-7B.

**Are harder tasks harder to forget?**   Another plausible explanation for why certain tasks are forgotten less is that harder tasks are more difficult to forget. However, as plotted in Figure 4, this is not consistently true. As a selected example, the forgetting procedure is less effective for the ARC easy dataset in comparison to the ARC challenge dataset, despite the significantly greater difficulty of the latter. Thus, the effectiveness of forgetting must be determined by other properties of the task.

**Does model confidence predict which tasks are forgotten?**   We hypothesize that a model's confidence may be predictive of whether a task is forgotten. The reasoning for this is that if the model has a strong preference for its answers on the task, a larger parameter update may be needed to overcome this "prior".

We examine the model's confidence in the correct response prior to running the forgetting procedure. Since the probability of the correct response is not calibrated, we measure the probability of the correct response relative to the incorrect response.

The results are shown in Figure 4. We find that the model's confidence in the correct response is partially predictive of how much the model forgets. Note that this is distinct from the difficulty of the task, as the model's confidence in the correct response is not necessarily correlated with whether it is actually correct.

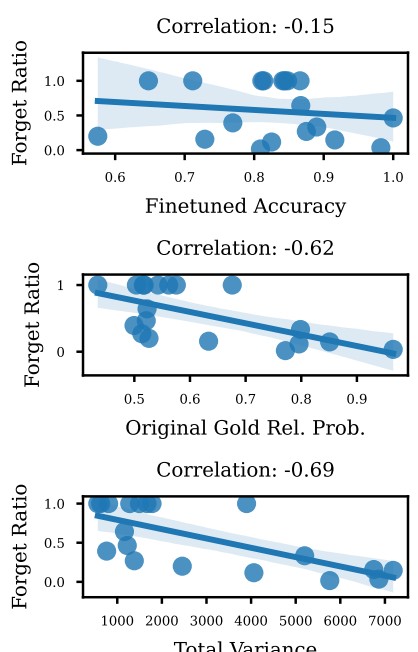

Figure 4: Predictors of the Forget Ratio (y-axis). Each point is a different task. *Top*: The accuracy on the task after fine-tuning. The effectiveness of the forgetting procedure is not determined by the difficulty of the task (as measured by accuracy). *Middle*: The variance of the hidden state of the last token of the question in the fifth to last layer across examples. This variance is somewhat predictive of amount forgotten, indicating that "broader" tasks are more difficult to forget. *Bottom*: Model's confidence in the correct response. Probability relative to the distractor is predictive of forgetting, indicating that models forget more examples they were already not confident about.

**Does hidden state variance predict forgetting?**   We also hypothesize that "broader" tasks are harder to forget. Since similar text is often mapped to similar regions in the latent space (Zhang et al., 2020), we use the variance of the hidden states of the model to quantify how much the model is able to forget. Specifically, we extract the hidden states at the last token of the question at the penultimate layer. We find that the total variance (trace of the covariance matrix) is predictive of how much the model is able to forget. Figure 4 shows that the smaller the total variance, the more effective the model is in forgetting. Note that this measure does not require access to the labels of the dataset, and it only requires access to the inference capabilities of the model and data from the task at hand.

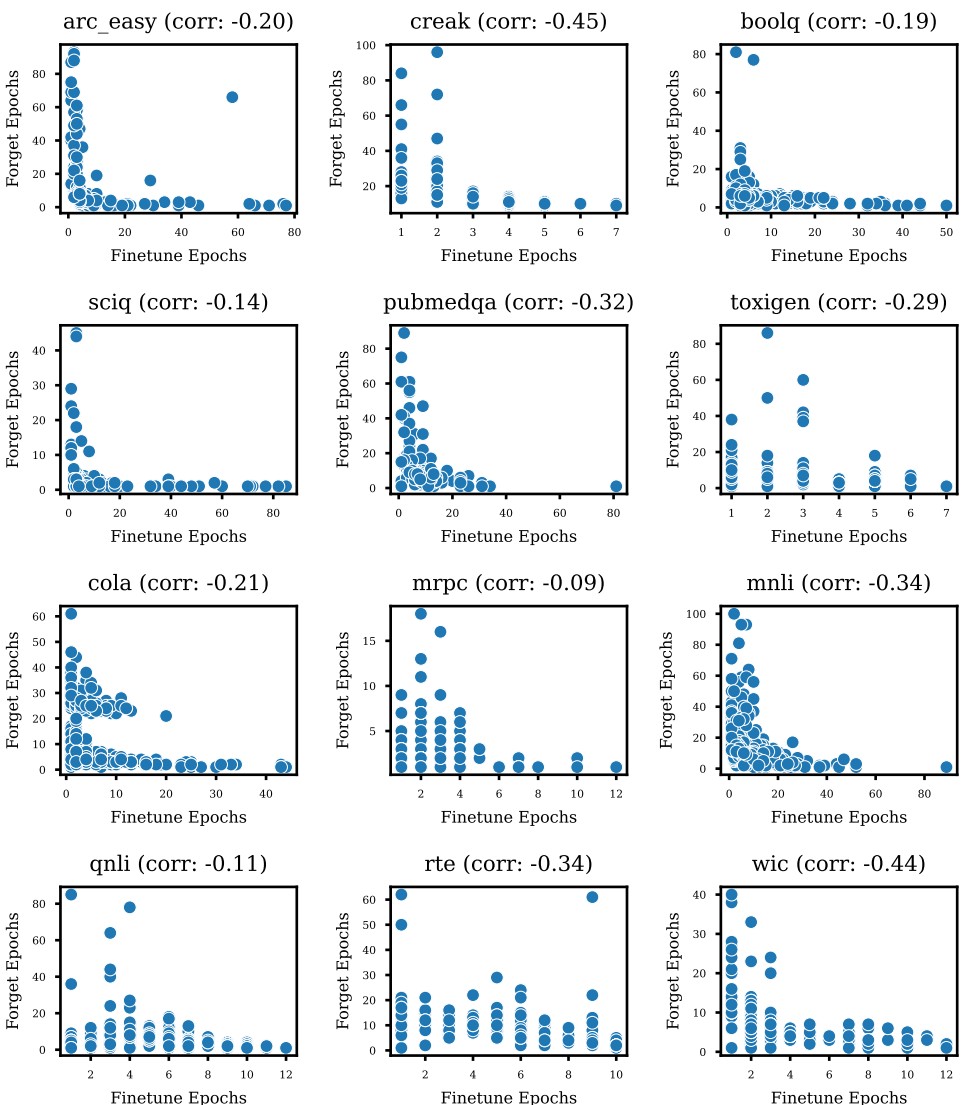

Figure 5: Forgetting order vs learning order. The horizontal axis shows the forgetting time: the number of epochs until the model forgets (assigns ¡ 60% accuracy to the correct response for a data point). The vertical axis shows the learning time: the number of epochs until the model learns (assigns ¿ 60% confidence to the correct label for a data point). We filter out the examples that are never learned or never forgotten. If fewer than 100 examples fulfil the criteria, we do not plot the task. Overall, we find that learning and forgetting orders are weakly, but consistently, anticorrelated.

**Can we predict which examples will be forgotten?** In contrast to the task-level trends depicted in Fig. 4, we did not observe any correlation between any of the above metrics and models' behavior at the level of individual examples—for example, example-level model confidence is not predictive of example-level forgetting. We hypothesize that different effects may dominate in this finer-grained scope, and that focusing on a narrow scope of same-task examples, other effects we did not yet uncover are too strong to see an effect with the current traits, such effects can be investigated in further work.

## 6    What is the relationship between learning and forgetting?

Even if extrinsic measures of difficulty cannot predict example-level learnability (as shown in the final experiment above), is there any systematic relationship between *learnability* and forgettability? Motivated by earlier work that similar architectures share consistent learning orders (Hacohen et al., 2020; Choshen et al., 2022), we hypothesize that the learning and forgetting *orders* are related.

As pre-trained models are often already partially capable of performing the tasks we study, we analyze learning orders after "resetting" the models to either extreme of the learning spectrum (maximum forgetting or maximum fine-tuning). Specifically, we compare the learning order of when we (1) run the forgetting procedure after fine-tuning (the same as in Section 4) and (2) when we run the fine-tuning procedure one more time afterwards (run fine-tuning after procedure in Section 4). Note that to prevent the models from learning all the examples in one epoch, we use a different fine-tuning learning rate of 3e-5 for experiment (2).

To qualify an example as learned, we require the model have a confidence of at least 0.6 in the correct response. To qualify an example as forgotten, we require the model have a confidence of at most 0.6 in the correct response. In preliminary experiments, we did not find results to be sensitive to the choice of threshold. For the purpose of analysis, we ignore examples that are never learned or forgotten. If no more than 100 examples fulfill the criteria, we do not plot the task. We take the first time this occurs as the forget time/learn time.

We visualize the learning orders in Figure 5. Across tasks, we find a consistent, modest correlation between learning order and forgetting order, in which the first points to be learned are typically the last to be forgotten and vice-versa. Overall, we hypothesize that the lack of a stronger correlation may be due to the shallow nature of fine-tuning. Since we are only aligning the model to the task instead of teaching it new capabilities, the learning order may be unaffected by example-level properties like difficulty. Thus, the learning order may be more related to the model's initial state.

## 7    Are "forgotten" skills truly removed from models?

One further question is if training on random labels really erases models' capabilities or if it only censors the output. To examine this, we train a linear probe on the models hidden states after performing the forgetting procedure. The probes are trained on the training set and evaluated on the test set. $\ell_2$ regularization and early stopping on a validation set used to prevent overfitting as the hidden state dimension is often larger than the number of examples. We select the fifth last layer of the model as the hidden state to probe, as we find that the accuracy of probing is mostly comparable for all layers except for the very early layers and the very late layers.

The results are shown in Figure 6. We find that the fine-tuning procedure largely does not influence the probing effectiveness. Thus, this procedure induces at best a shallow forgetting. This is consistent with most work that fine-tuning is often a shallow operation that does not significantly alter the model's capabilities (e.g.; Yadav et al., 2023; Horwitz et al., 2024).

## 8    Conclusion

In this paper, we study the effectiveness of fine-tuning models on randomized responses in order to forget capabilities. We find that this method is effective for certain tasks, but surprisingly does not generalize for others. The degree of forgetting seems mostly determined by the tasks that the model is evaluated on, not the tasks that the model was trained to forget. We find that dataset difficulty and model confidence are not predictive of whether a task is forgotten. However, we find that the total variance of the hidden states of the model is predictive of how much the model is able to forget. Finally, we show that despite the

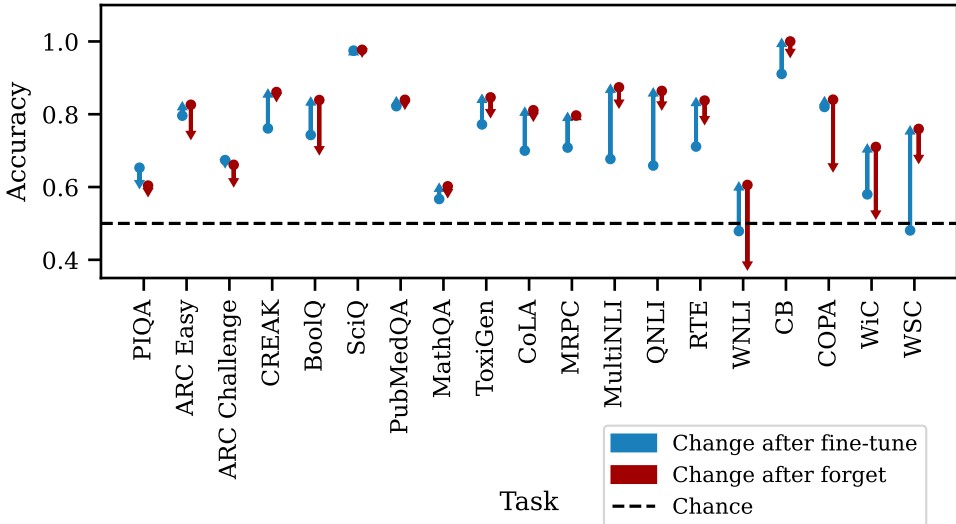

Figure 6: Probe accuracy. We plot the accuracy of a linear probe trained to classify (question, answer) pairs as correct or incorrect given LM hidden representations after pre-training, after fine-tuning, and after training on random labels. We find that forgetting largely does not influence the probing effectiveness, indicating that tasks are not truly forgotten even in cases where models generalizably learn to produce random outputs.

models' inability to respond correctly to prompts after applying this method, we are still able to recover the correct responses using linear probes. Thus, this is at best a shallow type of forgetting and not true removal of information from the model.

Future work can focus more on understanding which specific examples are forgotten and why. While our methods were successful in predicting which broad capabilities are forgotten, they are not predictive of which specific examples are forgotten within a task. This suggests that there are more mechanisms at play that can be studied further.

## Acknowledgments

This work was supported by the National Science Foundation under grant IIS-2238240. EZ is additionally supported by Liberty Mutual through the MIT Quest for Intelligence.

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

# A   Additional fine-tuning details

Unless otherwise stated, we perform full fine-tuning in half-precision with stochastic gradient descent and a learning rate of $3e-3$ with constant learning rate scheduling and gradient clipping of 1. Initial results with Adam were similar but required more memory. We fine-tune for 100 epochs with early stopping based on validation accuracy. We use a batch size of 3 which was the largest batch size that would fit in V100's memory. We fine-tune only on the response and never on the prompt. We fine-tune for 100 epochs or until the training set reaches 99% accuracy.

For our forgetting procedure, we randomly select either the correct response or the distractor before fine-tuning the model on that response in each epoch. Since allowing arbitrarily large learning rates can always lead to forgetting, we selected a learning rate where forgetting occur gradually over multiple epochs, $1e-4$. To prevent undertraining, we run the forgetting procedure for 100 epochs or until the model's test accuracy drops below 50%, whichever comes first.

# B   Reduced dataset size

| Task | Small Forget Accuracy | Large Forget Accuracy |
|---|---|---|
| PIQA | 0.71 | 0.69 |
| ARC Easy | 0.84 | 0.86 |
| ARC Challenge | 0.66 | 0.50 |
| CREAK | 0.71 | 0.77 |
| BoolQ | 0.77 | 0.50 |
| SciQ | 0.84 | 0.76 |
| PubMedQA | 0.73 | 0.63 |
| MathQA | 0.52 | 0.56 |
| ToxiGen | 0.80 | 0.77 |
| CoLA | 0.59 | 0.50 |
| MRPC | 0.77 | 0.80 |
| MultiNLI | 0.61 | 0.79 |
| QNLI | 0.71 | 0.50 |
| RTE | 0.57 | 0.50 |
| WNLI | 0.97 | 0.97 |
| CB | 0.61 | 0.50 |
| COPA | 0.51 | 0.50 |
| WiC | 0.62 | 0.50 |
| WSC | 0.63 | 0.66 |

Figure 7: Small dataset forgetting. To explore whether we have enough sample points for forgetting, we also run an experiment where only 100 examples are used for forgetting instead of 1000 in the large setting. We find that certain datasets exhibit less forgetting with the smaller dataset. However, the general trends remain the same, showing that the problem is not due explained fully by dataset size.

## C Flipped-label task

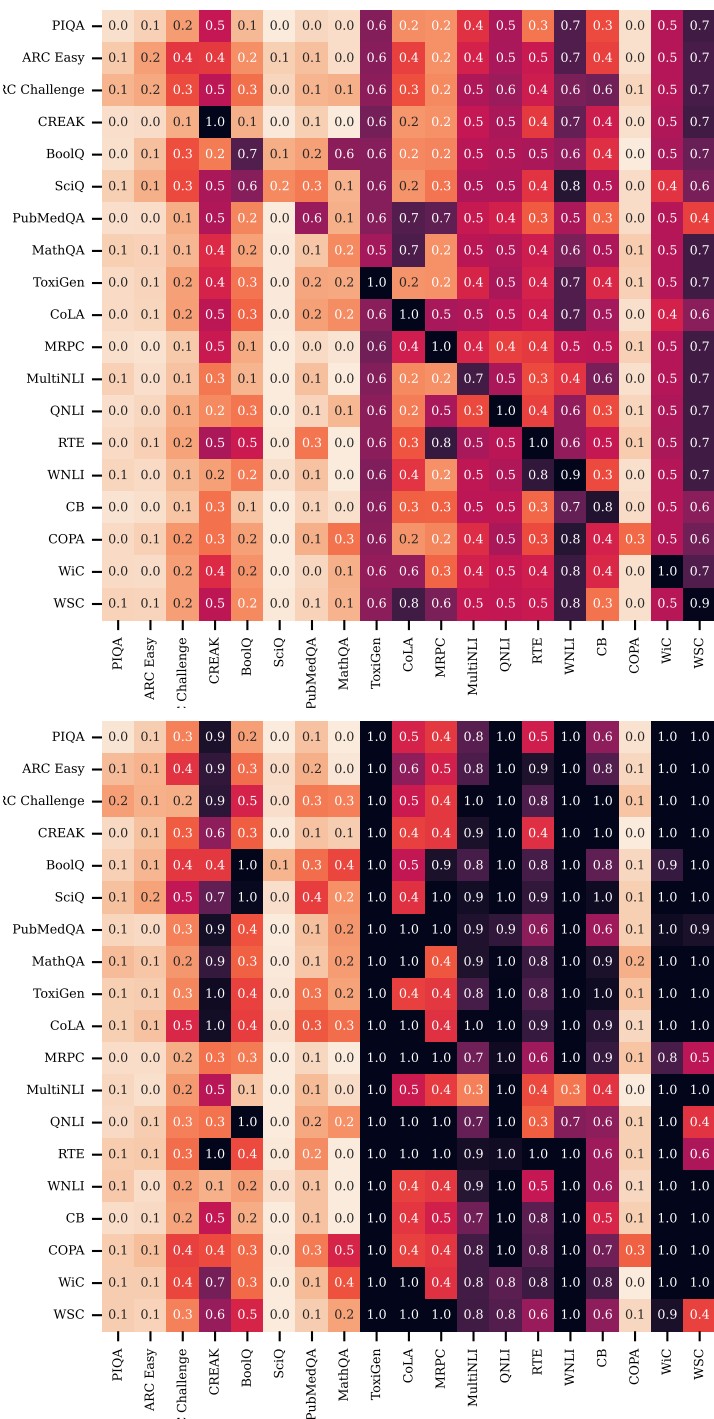

Figure 8: Flipped label forgetting. *Top*: Forget ratios for forgetting on the flipped task (higher values indicate more successful forgetting) vs *Bottom*: Forget ratios on the randomized task. We fine-tune the model on random/flipped labels from one task and then evaluate the model on another task. The vertical axis displays the task the model was trained to forget and the horizontal axis displays the task the model was evaluated on. We see similar trends in forgetting generalization in both task constructions.

## D   Other language models

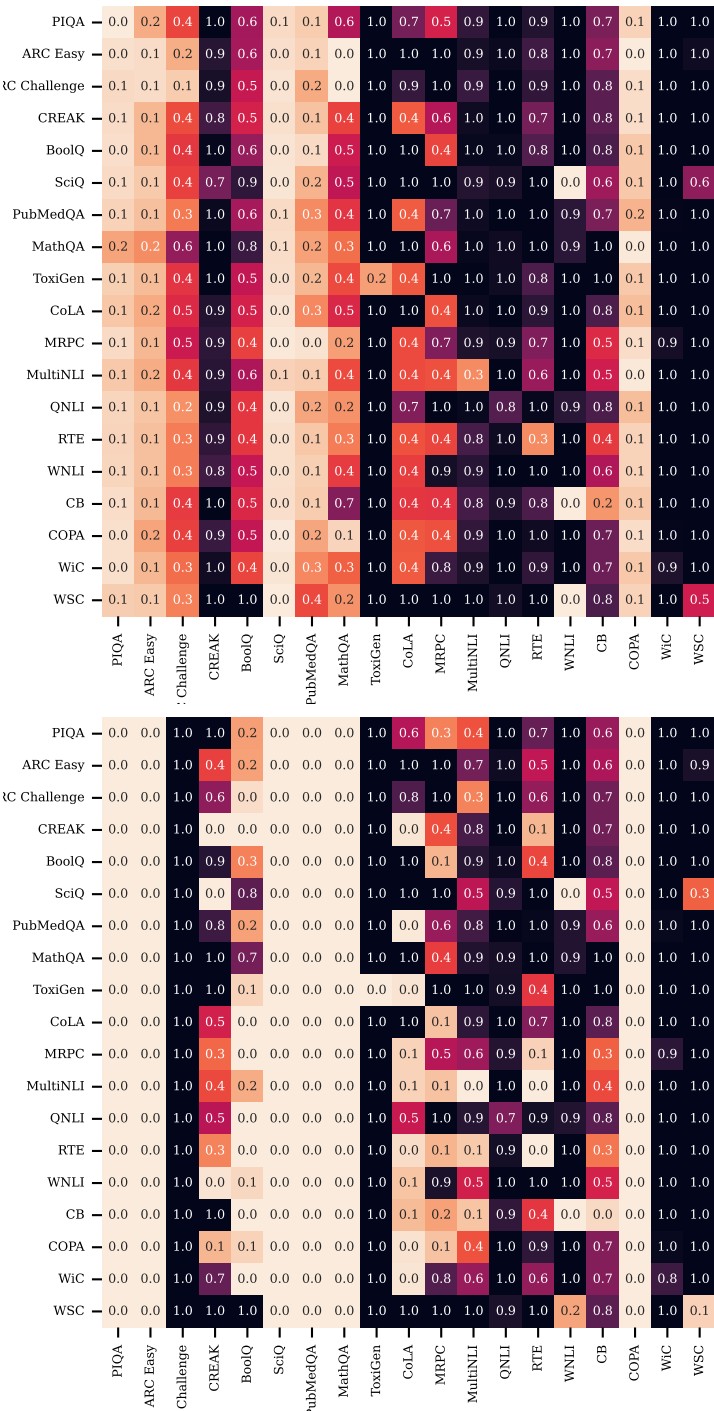

Figure 9: Forgetting performance of other models. *Top*: Forget ratios for cross task on the randomized task for GPT-J-6B (higher values indicate more successful forgetting) vs *Bottom*: Forget ratios for cross-task forgetting on the randomized task for GPT-2. The vertical axis displays the task the model was trained to forget and the horizontal axis displays the task the model was evaluated on. We see similar trends in forgetting generalization in both models and also the Llama2-7B model, which is shown in the bottom of Figure 8.

