# OpenReview forum: "Unforgettable Generalization in Language Models"
_colmweb.org/COLM/2024/Conference — COLM_

### Official Review · Reviewer_Leia · 2024-05-10

**Rating:** 7
**Confidence:** 2
**Ethics Flag:** 1

**Summary:**

The paper studies how "unlearning" of LM capabilities affects LM generalization. The main procedure is simply to finetune a LM on a target task and then fine-tune again with the labels being randomized. . The authors demonstrate with a series of analyses that this random-label unlearning procedure has variance in its effectiveness across tasks. The authors also show that unlearning does not in fact completely remove all information learnt during pre-training. It is possible to still use some linear probes to get correct responses.

**Questions To Authors:**

1. Can you explain the impact of the choice of random labels on your experiments in more details?

**Reasons To Accept:**

1. The contribution is novel.
2. The work explores both upsides and downsides of unlearning in significant detail. The authors do run an exhaustive series of experiments to demonstrate their claims.

**Reasons To Reject:**

1. It is unclear how the choice of random labels during the second phase of finetuning impacts the results. From my understanding of the paper, the labels are randomized. How different is this random label distribution from the original label distribution? What happens if the majority of labels match the original labels? Also, I would also like to know how the much variance is there in the unlearning process because of this randomness.
1. Some of the discussions about the experiments are vague/ambiguous. It is really hard to understand what the author means by a LM's "forget capabilities" or "unlearning of skills". It would be nice to explain these more in the paper or use different wording completely that is clearer.

---

> ### Author Rebuttal · Authors · 2024-05-30
>
> Thank you for your insightful comments and suggestions. We will revise our paper as follows.
>
> **"How different is this random label distribution from the original label distribution? What happens if the majority of labels match the original labels? Also, I would also like to know how the much variance is there in the unlearning process because of this randomness."**
>
> When preparing our datasets, we binarized our multiple-choice questions to always have two answers: the correct answer and a distractor. Thus, when we randomize our labels each epoch, there is a 50\% chance of it being the correct label. Note that the effect of randomizing our labels every epoch when combined with the fact that we are training on about 1000 examples for each task means that it is highly unlikely that a significant majority of labels match the original labels. The variance of label selection really is not that significant given all this repeated randomization.
>
> To demonstrate this, we also have a new experiment where we train on flipped labels instead of using randomized labels. Thus, in this setup, there is no variance due to randomized label selection.
>
> The analysis is the same, except now we compute the forget ratio as:
> $$\frac{\text{Fine-tuned Accuracy} - \text{Forgotten Accuracy}}{\text{Fine-tuned Accuracy} - (1- \text{Fine-tuned Accuracy})}$$
> since we assume that the minimum accuracy achievable should be $1- \text{Fine-tuned Accuracy}$.
>
> As shown in the [figure](https://ibb.co/P6Vp038), the trends are the same. The same tasks that are robust to fine-tuning on randomized labels continue to be robust on fine-tuning on flipped labels. Thus, our results are not specific to the choice of randomized labels, but are rather a property of how fine-tuning and the tasks interact.
>
> **"Some of the discussions about the experiments are vague/ambiguous. It is really hard to understand what the author means by a LM's "forget capabilities" or "unlearning of skills". It would be nice to explain these more in the paper or use different wording completely that is clearer."**
>
> Thank you for pointing this out. We will revise our paper and explain those terms. Specifically in regards to "forget capabilities" and "unlearning of skills," we use these interchangeably. In practice these refer to the out-of-sample performance on the tasks degrading--- which is what we call forgetting. We will be more careful with defining these terms and using them consistently.

---

> > ### Comment · Reviewer_Leia · 2024-06-03
> >
> > Thanks for the detailed response. I have decided to go with my original score. I would encourage the authors to include these clarifications in the revised version of the paper.

---

### Official Review · Reviewer_1y8m · 2024-05-10

**Rating:** 6
**Confidence:** 3
**Ethics Flag:** 1

**Summary:**

This work presents an analysis of targeted forgetting in LLMs via random-label fine-tuning. Random-label fine-tuning is a previously proposed approach to train an LLM to "unlearn" certain tasks: this work presents an analysis of when and why random-label fine-tuning actually results in "unlearning" of a task beyond the data that the model is trained to forget.

The authors fine-tune an a Llama2 7B LLM on 21 NLP tasks with their true labels and then fine-tune the resulting model on those tasks with random-labels. They show that not all tasks are effectively unlearned by random-label fine-tuning: while some tasks achieve random-chance on test data after unlearning, other tasks still generalize well to their test data. They also evaluate "cross-task forgetting", where they analyze the impact of unlearning a task on the generalization of other tasks; here they find, surprisingly, that many tasks are consistently forgotten during unlearning, regardless of the task being unlearned.

The authors then analyze why certain tasks are easily forgotten when others are not. They rule out several possibilities, showing low correlation between forgetting and task difficulty (measured as gold-label fine-tuned accuracy) and that more data does not improve forgetting. They instead find that a model's confidence (the probability of the correct response to the incorrect response) and the variance of the penultimate representations for each task are similarly correlated with forgetting (less confident tasks are easier to forget and tasks with similar representations across inputs are easier to forget).

Next, the authors study the order in which samples are learned vs. unlearned. They show that there is a loose relationship here, finding that samples which are learned later are forgotten earlier during fine-tuning, although samples that are learned early can also be forgotten early.

Finally, the authors present an analysis of how forgetting can affect the information stored within the intermediate representations of a model by training linear probes before and after training to forget. They find that, generally, training a model to forget a task does not significantly diminish the information of that task in the intermediate layers, as linear probes can often still generalize well to tasks after forgetting has occurred.

**Reasons To Accept:**

Targeted forgetting is an important topic in LLMs that is still poorly understood and this work takes steps in improving our understanding of it. The main observation that fine-tuning on random labels does not always generalize well across tasks but instead that certain tasks are almost always forgotten during cross-task forgetting is an interesting and important observation.

The work then presents a series of broad analyses that individually improve our understanding of when and why certain tasks are forgotten over others, and what the underlying mechanisms behind forgetting might be.

**Reasons To Reject:**

While the initial findings on the generalization of unlearning are interesting and the breadth of analysis of the rest of the work is impressive, I find the final sections (5-7) to be somewhat shallow individually.

More specifically, I think the work makes several interesting observations (certain tasks are always forgotten during unlearning regardless of target task, certain factors are somewhat correlated with unlearning generalization, hard samples are forgotten first, task-specific information is still retained in the layers). However, there is not a cohesive take-away from this analysis, i.e. it is not clear how this information should be used to improve targeted forgetting. I feel that the work would be improved if a depth-first approach to the analysis had been taken, i.e. if more resources had been devoted to understanding why certain tasks are always forgotten, or using the analysis in section 5 to present interventions that could improve forgetting.

As it stands, I think the work is weakened by the fact that the series of findings in 5-7 do not have a cohesive and clear impact on our ability to improve forgetting.

---

> ### Author Rebuttal · Authors · 2024-05-30
>
> Thank you for your insightful comments and suggestions. We will revise our paper as follows.
>
> **"I feel that the work would be improved if a depth-first approach to the analysis had been taken, i.e. if more resources had been devoted to understanding why certain tasks are always forgotten"**
>
> We agree that there is more work required to understand why certain tasks are always forgotten when other tasks are not. We did try other more bespoke approaches (including analyzing distances in representation space) but found no more success than in the simple factors already presented.
>
> As you have stated, our main observation is that
> > fine-tuning on random labels does not always generalize well across tasks but instead that certain tasks are almost always forgotten during cross-task forgetting is an interesting and important observation.
>
> Our main contribution is observing this and considering immediate explanations that might come to mind. This intriguing finding definitely raises more questions for follow-up work.

---

> > ### Comment · Reviewer_1y8m · 2024-06-05
> > **Rebuttal Response**
> >
> > Thank you for your response. I definitely agree that there is a lot of work still to do to understand this area, and I appreciate that this work makes a meaningful contribution to our understanding with it's core observation.
> >
> > I still feel as though the paper overall lacks a cohesive takeaway throughout the later sections and so I will keep my score the same (leaning positive because I do feel that the main finding outweighs the negatives).

---

### Official Review · Reviewer_Rju9 · 2024-05-10

**Rating:** 7
**Confidence:** 3
**Ethics Flag:** 1

**Summary:**

This paper explores unlearning of skills for language models. Specifically they explore what is the extent to which the unlearning procedure of finetuning w/ random label generalizes. They find that generalization is very task dependent but surprisingly does not depend on what dataset was used for the unlearning procedure. The paper then analyzed some factors which could be helpful to predict what can or cannot be unlearned — dataset difficulty, model confidence, and how ‘broad’ the task is. Lastly the paper shows that unlearning even if it generalizes is still shallow – we can train linear probes for the skill very easily.

**Questions To Authors:**

1. The figure 3 findings are pretty interesting! Do you have a hypothesis/guess for why it doesn't matter what dataset or task you use for unlearning?

2. Section 6 – how was the threshold of 0.6 determined?

3. Conclusion second para(typo?) – “methods were not successful” → “methods were successful”?

4. Section 5 – can you add the actual results for training with 100 examples? (maybe in the appendix)

5. Section 5 / figure 4 – it would be useful to state explicitly that you use finetuned accuracy as a measure of the hardness of the task.

**Reasons To Accept:**

1. The problem studied seems quite important and relevant! Often in areas like knowledge editing people look at the specificity of the editing procedure — this paper explores a similar idea but for unlearning.

2. The datasets used seem quite exhaustive, and the evaluation setup seems pretty standard (LM evaluation harness) which is good.

3. The paper has some interesting findings which I believe could be both impactful and practically useful.

    a. They show that there is sometimes asymmetry between learning and forgetting process (figure 2 top)

    b. Figure 3 – It shows that it doesn't really matter what task/dataset we use for unlearning – the effects of unlearning on different dataset remain the same. This is very surprising and counterintuitive.

    c. Figure 4 – The overall finding that model confidence and how niche/broad the task (as measured by hidden state variance) are correlated with forget ratio is also pretty interesting.

4. The paper is well written, and I think all result figures seem thorough and informative.

**Reasons To Reject:**

1. While the datasets explored are quite exhaustive, the methods for unlearning and the models explored are very limited. The paper only looks at one method (training w/ random labels) but there are a lot of other methods proposed in the literature; but more importantly only one model Llama2-7B is explored — it’s unclear if the findings are more general or specific to this model. I do think the paper could be much stronger by adding some more models in the analysis.

2. The discussion of prior work and relevant literature is somewhat lacking

    a. It seems like other papers have studied generalization of unlearning –e.g. https://arxiv.org/abs/2403.03218 shows that their proposed unlearning method can maintain performance on other domains (e.g. biology, computer science). A more detailed discussion and comparison to prior work would be useful.

    b. (minor) There are also works from slightly different domains which seem relevant and could be included here — generalization of the editing/unlearning procedure (also called as ‘ripple effects’) in some papers – https://arxiv.org/abs/2307.12976, https://arxiv.org/abs/2305.01651 etc. ; methods like iterative null space projection (https://arxiv.org/abs/2004.07667) and follow-ups.

---

> ### Author Rebuttal · Authors · 2024-05-30
>
> Thank you for your insightful comments and suggestions. We will revise our paper as follows.
>
> **"The discussion of prior work and relevant literature is somewhat lacking."**
>
> We took the suggestion to expand prior work to heart and will do so. For arxiv.org/abs/2403.03218, they use the same term "generalization" but study a different kind of generalization--- not breaking unrelated tasks. Our paper focuses on generalization within a single task. We will add this discussion.
>
> **"The paper only looks at one method (training w/ random labels)."**
>
> Instead of studying generalization in all possible unlearning methods, we focus on fine-tuning which is simple and a common baseline.
>
> Additionally, fine-tuning is especially interesting because it may reveal connections between learning and unlearning (e.g. there seems to be wide vs narrow tasks, which we imagine would have consequences in learning).
>
> To further address this, we replicated our results with a new experiment using flipped labels instead of randomized labels. The analysis is the same, except now we compute the forget ratio as:
> $$\frac{\text{Fine-tuned Accuracy} - \text{Forgotten Accuracy}}{\text{Fine-tuned Accuracy} - (1- \text{Fine-tuned Accuracy})}$$
> since we assume that the minimum accuracy achievable should be $1- \text{Fine-tuned Accuracy}$.
>
> As shown in the figure ibb.co/P6Vp038, forgetting trends are the same. Our results are not specific to the choice of randomized labels.
>
> **"Only one model."**
>
> We conducted a new experiment with GPT-J-6B. As shown in ibb.co/pRfwByd, the forgetting ratio trends are broadly the same, so the behavior is not unique to LLaMA-2-7B.
>
> Moreover, to test what happens to models of different sizes, we also run a new experiment on GPT-2 with 124M parameters. As shown in ibb.co/TK9x7j4, the forgetting ratio trends still remain the same.
>
> **"Do you have a hypothesis/guess for why it doesn't matter what dataset or task you use?"**
>
> We suspect that certain capabilities are inherently more fragile, so fine-tuning on any incorrect data is harmful. This resonates with recent findings that highly quality datasets are important.
>
> **"How was the threshold of 0.6 determined?"**
>
> We picked a threshold of 0.6 which is between 0.5 and the original confidence. The experiments are robust to thresholds in that neighborhood.
>
> **Typos, clarifications, and adding data**
>
> Thank you. We will amend the typos, make the suggested clarifications, and add the requested results to the appendix.

---

> > ### Comment · Reviewer_Rju9 · 2024-06-06
> > **Response to Rebuttal**
> >
> > Thanks for responding to the review! I think the authors have satisfactorily addressed the concerns I had --- specifically they ran new experiments with one more model, as well as a slightly different unlearning method (flipped label). I hope the authors add more discussion about related work as they mentioned in the rebuttal.
> >
> > Considering this, I'm happy to increase the score.

---

### Official Review · Reviewer_7Lpc · 2024-05-12

**Rating:** 6
**Confidence:** 4
**Ethics Flag:** 1

**Summary:**

The paper investigates the unlearning process in language models by fine-tuning on randomized labels. The paper is presented clearly with a systematic analysis of the factors affecting the generalization of unlearning. The results show that the effectiveness of unlearning is task-dependent and if often shallow. However, the analysis is limited to random-label unlearning, and the reasons behind the generalization of unlearning are not well-explained.

**Questions To Authors:**

* Would other unlearning methods yield different results in terms of generalization? For example, using label-flipping, label transformation, or label of a different task on the same inputs?

* Would the size of the model affect the generalization of unlearning? Also, extending the experiments to another language model besides LLaMA might provide more insights into the generalizability of unlearning.

**Reasons To Accept:**

* The paper presents a novel investigation into the unlearning process in language models. The results show that the effect of unlearning can be task-dependent and can be uneffective for many tasks, which calls for a deeper understanding of the unlearning process and more effective methods for unlearning with language models.

* The paper is well-written, with a clear structure and presentation of the experimental results. The authors present a comprehensive analysis of multiple factors that could affect the generalization of unlearning, such as dataset difficulty, model confidence, representation variability, and learning orders.

**Reasons To Reject:**

* Method of unlearning is limited to fine-tuning with randomized labels, and it is unclear from the paper whether the failure of generalization of unlearning is due to the unlearning method being ineffective or due to the nature of the language model and the tasks.

* Absent of clear explanation on why unlearning generalizes for some tasks and not for others. The paper claims that the effectiveness of unlearning generalization is task-dependent, but the reasons behind this are not well-explained. This limits the practical implications of the findings.

---

> ### Author Rebuttal · Authors · 2024-05-30
>
> Thank you for your insightful comments and suggestions. We will revise our paper as follows.
>
> **Method of unlearning is limited to fine-tuning with randomized labels.**
>
> Following your question, we ran another experiment where we used flipped labels instead of randomized labels. The analysis is the same, except now we compute the forget ratio as:
> $$\frac{\text{Fine-tuned Accuracy} - \text{Forgotten Accuracy}}{\text{Fine-tuned Accuracy} - (1- \text{Fine-tuned Accuracy})}$$
> since we assume that the minimum accuracy achievable should be $1- \text{Fine-tuned Accuracy}$.
>
> As shown in the [figure](https://ibb.co/P6Vp038), the trends are the same. The same tasks that are robust to fine-tuning on randomized labels continue to be robust on fine-tuning on flipped labels. Thus, our results are not specific to the choice of randomized labels, but are rather a property of how fine-tuning and the tasks interact.
>
> **Extending the experiments to another language model besides LLaMA might prove more insights into the generalizability of unlearning.**
>
> We conducted a new experiment with GPT-J-6B, which is a slightly weaker model than the LLaMA-2-7B model we used. As shown in the [figure](https://ibb.co/pRfwByd), while GPT-J has a lower fine-tuned accuracy, the forgetting ratio trends are broadly the same. Thus, the behavior is not unique to LLaMA-2-7B.
>
> Moreover, to explicitly test what happens to models of different sizes, we also run a new experiment on GPT-2, which has 124M parameters (98\% smaller). As shown in the [figure](https://ibb.co/TK9x7j4), GPT-2 tends to have significantly lower fine-tuned accuracy, but the forgetting ratio trends are broadly the same even in the smaller model size regime.
>
> We will add these experiments and their discussion.
>
> As the generality of our findings was judged to be the main weakness, we hope this cleared the doubts about the paper.

---

> > ### Comment · Reviewer_7Lpc · 2024-06-05
> >
> > Thanks for your response. The newly added results seems convincing and addresses my concerns, therefore, I have improved the score. I look forward to see a more comprehensive discussion with the expanded results in the revision of the paper.

---

### Decision · Program_Chairs · 2024-07-10

**Decision:**

Accept

**Comment:**

This paper studies unlearning in LMs, whereby an LM is "finetuned" on a task with randomized labels. The paper empirically studies the effects of this procedure, especially, how forgetting does or does not generalize across tasks. The findings are likely to have implications for model editing and targeted forgetting, an important topic in LLMs.

Reviewers were all positive. Summarizing my reading of the reviews and the papers, and taking the author responses into account (which have ameliorated some issues), I believe the following are relevant:

Strengths:
- novel (7Lpc) and important (Rju9, 1y8m, Leia)
- interesting empirical results (Rju9, 1y8m, Leia)

Weaknesses:
- no explanation of why the effect of unlearning depends on tasks (7Lpc) and, as a consequence, unclear take-aways (1y8m)

Taken together, I concur with the reviewers that this is a novel approach to studying the effects of model editing, an important topic for today's LMs. While the observed phenomena are not explained in the paper, I believe the paper still has sufficiently interesting and rigorous empirical findings.